# ACCTS: AN ADAPTIVE MODEL TRAINING POLICY FOR CONTINUOUS CLASSIFICATION OF TIME SERIES

## ABSTRACT

More and more real-world applications require to classify time series at every time. For example, critical patients should be detected for vital signs and diagnosed at all times to facilitate timely life-saving. For this demand, we propose a new concept, Continuous Classification of Time Series (CCTS), to achieve the high-accuracy classification at every time. Time series always evolves dynamically, changing features introducing the multi-distribution form. Thus, different from the existing one-shot classification, the key of CCTS is to model multiple distributions simultaneously. However, most models are hard to achieve it due to their independent identically distributed premise. If a model learns a new distribution, it will likely forget old ones. And if a model repeatedly learns similar data, it will likely be overfitted. Thus, two main problems are the catastrophic forgetting and the over fitting. In this work, we define CCTS as a continual learning task with the unclear distribution division. But different divisions differently affect two problems and a fixed division rule may become invalid as time series evolves. In order to overcome two main problems and finally achieve CCTS, we propose a novel Adaptive model training policy - ACCTS. Its adaptability represents in two aspects: (1) Adaptive multi-distribution extraction policy. Instead of the fixed rules and the prior knowledge, ACCTS extracts data distributions adaptive to the time series evolution and the model change; (2) Adaptive importance-based replay policy. Instead of reviewing all old distributions, ACCTS only replays the important samples adaptive to the contribution of data to the model. Experiments on four real-world datasets show that our method can classify more accurately than all baselines at every time.

## 1 INTRODUCTION

In the real world, many applications need to classify time series data at every time (Gupta et al. (2020)). For example, in the Intensive Care Unit (ICU), most detected vital signs of patients change dynamically according to the development of decease. The status perception is needed at any time as the real-time diagnosis provides more opportunities for doctors to rescue lives (Chen et al. (2014)). In response to the current demand, we propose a new concept – Continuous Classification of Time Series (CCTS). It aims to classify as accurately as possible st every time with the data evolement.

For most time series extracted from practical applications, the development, caused by their changed data characteristics, leads to the evolved data distribution, and finally produces the multi-distribution form. For example, in Figure 1, the data distributions of blood pressure of 2,000 sepsis patients vary among early, middle and late time stages during hospitalization, bring a triple-distribution.

In the background of CCTS, modeling multiple data distributions simultaneously is the requirement: when the data distribution changes, the model performance cannot decrease. However, a single model, like deep neural network, is lack of ability to learn all distributions simultaneously as they are restricted by the premise of independent identically distributed (i.i.d) data (Shim et al. (2021)). If a model learns a new distribution, it will negatively affect its performance on old ones.

Therefore, when learning the multi-distribution, a model will face two problems:

- Catastrophic forgetting. A time series usually has a large number of time points. For example, the blood pressure of a critical patient could be sampled hundreds of times. If a model is trained at all points, it will learn hundreds of data distributions continuously. Frequent learning of new knowledge will inevitably lead to the forgetting of old ones (Parisi et al. (2019));

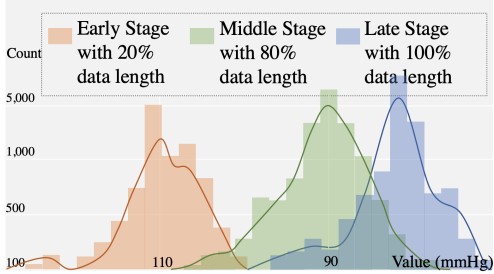 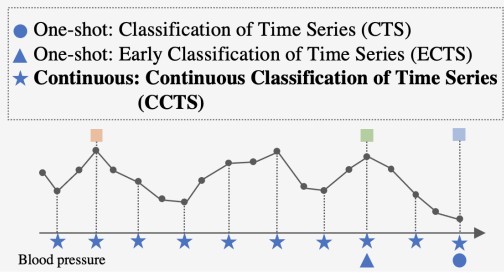

Figure 1: Multi-distribution in Time Series. The statistics of mean blood pressure of 2,000 sepsis represent the distributions (Kiyasseh et al. (2021)).

Figure 2: Continuous Classification Mode. Different from the existing time series classification task, CCTS need the continuous classification.

- Over fitting. As the development of time series needs a process, the data distributions of adjacent time are always similar as shown in Figure 1. And in order to alleviate the catastrophic forgetting, many methods replay old distributions to enhance memory (Kiyasseh et al. (2021)). Over-learning of similar data will cause the strict function and poor generalization (Saha et al. (2021)).

Two issues are intertwined. The old knowledge review could avoid the forgetting but worsen the overfitting, while the iteration avoidance could alleviate the overfitting but worsen the forgetting. The additional policy like the selective review, could help to balance the above problems. However, the policy formulation is based on the known data distribution, yet the distribution in CCTS is not clear as its data form is sequential and can be divided by different rules. Therefore, before planning how to learn the multi-distribution, we should also determine the well-divided multi-distribution as:

- Less divided distributions may worsen the catastrophic forgetting problem and omit important data features. If dividing time series less, distributions will be more different. It will make the model change greatly after learning the new distribution and seriously affect the model performance on old distributions. Besides, the key distribution may not be included due to the incomplete sampling;

- More divided distributions may worsen the over fitting problem and have the low training efficiency. If dividing time series more, distributions will have many overlaps. It will make some similar features to be learned repeatedly and over fit the trained model to the recurring dataset. Besides, in the model training phrase, learning more distributions will spend more time.

The optimal multi-distribution is hard to obtain. Unlike images, the time series is more abstract and its characteristics are not explicit (Xing et al. (2011)). Although some methods can describe time series like Shaplets (Liang & Wang (2021)), they still need the prior knowledge. Most importantly, the artificial rule needs to be determined before training the model and remains the same over time. But because the time series has been evolving dynamically, a fixed rule is likely to be outdated.

In this work, instead of the fixed methods, we design an Adaptive model training policy for Continuous Classification of Time Series (ACCTS). It has two data-based adaptive policy:

- Adaptive multi-distribution extraction policy. It explores the policy space according to the reward of distribution difference and classification accuracy based on the reinforcement learning strategy, and finally extracts data distributions adaptive to the time series evaluation and the model change;

- Adaptive importance-based replay policy. It leans the impact of each sample on the model, applying partial replay to balance the result accuracy and the time efficiency. The important samples in each distributions are obtained adaptive to the time series evaluation and the model change.

Both polices are dynamic rather than static to trade off the forgetting and the overfitting. Experimental results on real-world datasets show that ACCTS is more accurate than all baselines at every time.

## 2 RELATED WORK

The popularity of time series classification has attracted increasing attention in many practical fields (Santos & Kern (2016)). The foundation is Classification of Time Series (CTS), making classification based on the full-length data (Fawaz et al. (2019)). But in time-sensitive applications, Early Classification of Time Series (ECTS) is more critical, making classification at an early time (Gupta et al. (2020)). For example, early diagnosis helps for sepsis outcomes (Liu et al. (2018a)). Both of them give one classification result, while CCTS needs continuous multiple results. Thus, based on the classification mode, the existing work can be summarized into two categories.

## 2.1 ONE-SHOT CLASSIFICATION

This mode classifies time series at a fixed time: as shown in Figure 2, once the classification is complete, the action will not continue. Both CTS and ECTS are belong to this and always use Deep Learning (DL) models. RNNs-based methods recur in evolution direction to learn the sequential dependency (Fawaz et al. (2019)). CNN-based methods use kernels to extract local features (Huang et al. (2017)). Choi et al. (2017) models the long-term dependencies; Tan et al. (2020) and Sun et al. (2021) address the time irregularity; Hsu et al. (2019) learns frequency features; Lai et al. (2015) designs Temporal Convolutional Network (TCN) to integrate RNN and CNN. However, the above DL-based methods just perform well on i.i.d data at a fixed time, like early 6 hours sepsis diagnosis (Reyna et al. (2019a)), but fail for classifying old distributions after learning new distributions.

## 2.2 CONTINUOUS CLASSIFICATION

This mode classifies time series at every time as shown in Figure 2. Most of the existing methods apply multi-models to model multi-distribution, like SR (Mori et al. (2018)) and ECEC (Lv et al. (2019)). They divide data according to time stages and design different classifiers for different distributions. But they only consider the data division, ignoring the strategic training method. Besides, the operation of classifier selection in multi-models framework will result in additional losses.

CCTS has the multi-distributed data, leading to the catastrophic forgetting problem. Recently, Continual Learning (CL) (Delange et al. (2021)) aims to address the issue of static models incapable of adapting their behavior for new knowledge. It learns a new task at every new moment and each new task corresponds to a new data distribution. Replay-based methods re-train the model by old data to consolidate memory (Rolnick et al. (2019); Kiyasseh et al. (2021); Isele & Cosgun (2018); Rebuffi et al. (2017)); Regularization-based methods restrain parameter update of neural networks to limit forgetting (Kirkpatrick et al. (2016); Lopez-Paz & Ranzato (2017); Liu et al. (2018b); Zhang et al. (2020)); Model-based methods change network structure or apply multiple models to response to different tasks (Fernando et al. (2017); Mallya & Lazebnik (2018)). But most of the above methods have the problems of storage limitation, distributions drifts and model overfitting. Most importantly, in CL, the definition of old and new tasks is clear and the division of distribution is fixed. But in CCTS, the distributions, that is, the tasks in CL, is not determined and need to be defined.

Thus, CCTS is a new proposed concept, there is no systematic analysis and corresponding solutions.

## 3 PROBLEM FORMULATION

**Definition 1 Continuous Classification.** *A time series $X = \{x_1, ...x_T\}$ having $T$ time points is labeled with a class $C \in \mathcal{C}$. Continuous classification tasks aim to classify the time series $X$ at every time $t, t = 1, ..., T$ with the minimum additive loss $\sum_{t=1}^{T} \mathcal{L}(f(X_{1:t}), C)$.*

Note that most of the existing time series classification tasks are belong to the one-shot classification mode, where they optimize the objective with a single minimum loss $\mathcal{L}(f(X_{1:t}), C)$. In continuous classification, the model need to learn multiple distributions, which is firstly proposed in CL field. A CL issue $\mathcal{T} = \{\mathcal{T}^1, \mathcal{T}^2, ..., \mathcal{T}^N\}$ has a sequence of $N$ tasks, learning a new task $T = (X, C)$ at every time. The goal is to control the statistical risk of all seen tasks $\sum_{n=1}^{N} \mathbb{E}_{(X^n, C^n)}[\mathcal{L}(f^n((X^n; \theta), C^n)]$. Based on this settings, we give the definition of CCTS:

**Definition 2 Continuous Classification of Time Series (CCTS).** *A time series $X = \{x_1, ...x_T\}$ has $T$ time points and is labeled with a class $C \in \mathcal{C}$. CCTS has a sequence of $N$ distributions $\mathcal{M} = \{\mathcal{M}^1, \mathcal{M}^2, ..., \mathcal{M}^N\}$. Each distribution $M^n$ is represented by the subsequence set $X_{1:t^n}$. The goal is the statistical risk control of classification with all distributions: $\sum_{n=1}^{N} \mathbb{E}_{\mathcal{M}^n}[\mathcal{L}(f^n(\mathcal{M}^t; \theta), C)]$ with the loss $\mathcal{L}$, the network function $f$ and parameters $\theta$ of the classification model. When the model $f^n$ is trained by the current distribution $\mathcal{M}^n$, its performance on all observed data cannot degrade:*

$$\min \mathcal{L}(f^n, \mathcal{M}^n)$$

$$subject\ to\ \frac{1}{t^n} \sum_{t=1}^{t^n} \mathcal{L}(f^n(X_{1:t}; \theta^n), C) \leq \frac{1}{t^{n-1}} \sum_{t=1}^{t^n} \mathcal{L}(f^{n-1}(X_{1:t}; \theta^{n-1}), C) \tag{1}$$

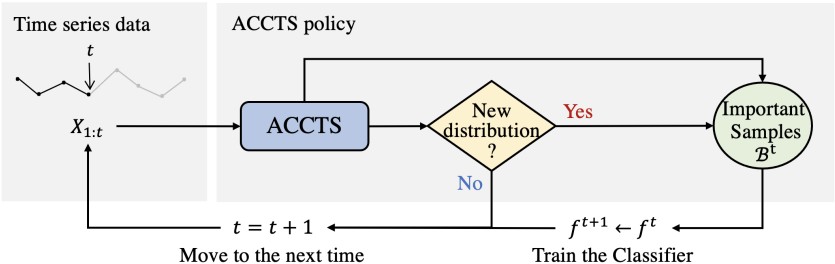

Figure 3: Adaptive Model Training Process for Continuous Classification of Time Series

## 4 ADAPTIVE MODEL TRAINING POLICY

Figure 3 shows the model training process by the policy of ACCTS. When a model is trained by time series from the initial to the final time, ACCTS gives two decisions:

- Whether the current time series segment forms a new distribution to be learned. If yes, train the model by the current time series; Otherwise, do not train and continue to get new data points;
- Which old samples need to be replayed and learned again. If the previous decision is yes, train the model by the obtained old samples again after train it by the current time series.

### 4.1 ADAPTIVE MULTI-DISTRIBUTION EXTRACTION

The first decision is got by the adaptive multi-distribution extraction polity. It is an agent that decides whether to extract the current time series sequence to train the model. It solves a 3-triple partially-observable Markov decision process $\{\mathcal{S}, \mathcal{A}, \mathcal{R}\}$ (Kaelbling et al. (1995)), where the observation arrive from a state $s$ at each time, an action $a$ is sampled using a learned policy, and a reward $r$ is observed according to the selected action's quality. The objective is to optimize long-term rewards.

**State $\mathcal{S}$.** It is represented by the characteristics of the currently data and the adaptability of the old model to the current data. It is intuitive: First, the model need to be trained by the dataset with different features from the previous data for the comprehensive modeling; Second, the model must be trained again when it performs poorly on the current data for the overall accuracy.

At the current time $t$, we use the Long Short-Term Memory (LSTM) network as the base model to learn the hidden characteristics of a time series $X_{1:t}$, generating low-dimensional vector representation $h_t$. We also propose the model gradients $g_t$ to evaluate the adaptability of the model to the current time series. The model gradient can help for the interpretation of DL model by explaining the response of the neural network to input data (Srinivas & Fleuret (2019)). Large gradient fluctuation reflects the low adaptability of the model to the input data. Thus, the state $s_t$ of current time series is:

$$s_t = \text{concatenate}(\text{LSTM}(x_t), \text{MG}(X_{1:t})) \tag{2}$$

$$\begin{aligned} f_t &= \sigma(W_f[h_{t-1}, x_t] + b_f), \ i_t = \sigma(W_i[h_{t-1}, x_t] + b_i), \\ c_t &= f_t \cdot c_{t-1} + i_t \cdot \eta(W_c[h_{t-1}, x_t] + b_c), \\ o_t &= \sigma(W_o[h_{t-1}, x_t] + b_o), \ h_t = o_t \cdot \eta(c_t). \end{aligned} \qquad \text{LSTM}$$

$$g_t = \frac{\partial}{\partial \theta_{f^n}} \mathcal{L}(f^n(X_{1:t}, \theta_f), C) \qquad \text{MG}$$

**Action $\mathcal{A}$.** At the current time $t$, the action $a_t$ dictates the decisions of ACCTS agent: If $a_t = 0$, continue to accept the value point of time series and let LSTM move forward one time step; If $a_t = 1$, extract the current time series $X_{1:t}$ as a new distribution to be learned. For the action selection, we use $\varepsilon$-greedy selection to avoid abundant exploitation. $a_t$ is replaced with a random action with the probability $\varepsilon$ of exponentially decreasing from 1 to 0 during the training process.

$$a_t = \begin{cases} a_t, & \text{with probability } 1 - \varepsilon \\ random, & \text{with probability } \varepsilon \end{cases}, \quad a_t \in \{0, 1\} \tag{3}$$

**Reward $\mathcal{R}$.** The agent observes the return which can qualify the parameters of the current policy. The goal of CCTS is the high accurate classification by solving the problems of catastrophic forgetting

---

**Algorithm 1** Adaptive Multi-distribution Extraction Policy

---

**Input:** Data $\mathcal{D} = \{(X, C)\}$;
      Classifier net $f$.

$\nabla_{\theta_\mathcal{Q}} O_{\text{critic}} = \sum_t (y - \nabla_{\theta_\mathcal{Q}} \mathcal{Q}(s_t, a_t | \theta_\mathcal{Q}))$

**Output:** Extraction policy $(\mu, Q)$.

$\nabla_{\theta_\mu} O_{\text{actor}} = \sum_t \nabla_{\mu(s_t)} Q(s_t, \mu(s_t)) \nabla_{\theta^\mu} \mu(s_t | \theta_\mu)$

  1: **for** $t = 1$ to $T$ **do**
  2:    $s_t \leftarrow$ Equation 2
  3:    $a_t \leftarrow \mu(s_t, \theta_{Actor})$
  4:    **if** $a_t = 1$ **then**
  5:       $r_t \leftarrow$ Equation 4
  6:       Update $\mathcal{Q}, \mu$ by:

  7:       Soft update $\mathcal{Q}', \mu'$ by:
           $\theta_{\mathcal{Q}'} \leftarrow \tau\theta_\mathcal{Q} + (1 - \tau)\theta_{\mathcal{Q}'}$
           $\theta_{\mu'} \leftarrow \tau\theta^\mu + (1 - \tau)\theta_{\mu'}$
  8:       Train $f$ by $(X_{1:t}, C)$
  9:    **end if**
10: **end for**

---

and over fitting as we analysed in Section 1. Thus, we pursue the higher accuracy of the current classifier on all potential data distributions to control the catastrophic forgetting, and we limit the number of extracted distributions by the time span between distributions to control the over-fitting. Thus, at the current time $t$, after applying the action $a_t$, the reward $r_t$ is consisted of two components. The first term is for the high accuracy of the current model $f^n$ on all data, the second term is for less divisions by using the time length between the current time $t^n$ and the last data extraction time $t^{n-1}$.

$$r_t = -\frac{\alpha}{T}\sum_{t=1}^{T}\mathcal{L}(f^n, X^{1:t}) + \frac{(1-\alpha)}{T}(|t^n - t^{n-1}|) \tag{4}$$

When using the transition probability $P(s_{t+1}|s_t, a_t)$, the total reward of the trajectory is the is sum of the reward in each time. Thus, the objective is to maximize the total reward $R = \sum_{t=1}^{T} r_t$. The policy gradient method (Sutton et al. (1999)) learns the policy $\pi_\theta(s_t, a_t) = P(a_t|s_t)$ for the larger return. The objective is $J(\theta) = \mathbb{E}[r(s, a)\pi_\theta(s, a)]$. For ACCTS, we apply Actor-Critic (Zhang et al. (2019)) structure with two components of the main net and the target net. The main net of Actor $\mu$ use the state $s$ to generate the action $a$; The main net of Critic $\mathcal{Q}$ judges the action $a$ through reward $r$ by Q-function (Watkins & Dayan (1992)). The target nets of Actor and Critic $\mu', \mathcal{Q}'$ put the target $Q$ value stable for a period of time, making the algorithm performance more stable.

$$O_{Actor}(\theta_\mu) = \mathbb{E}_{s_t \in S}[\mathcal{Q}(s_t, \mu(s_t|\theta_\mu)|\theta_\mathcal{Q})]$$
$$O_{Critic}(\theta_\mathcal{Q}) = \mathbb{E}_{s_t \in S}[(r_t + \gamma\mathcal{Q}'(s_{t+1}, \mu'(s_{t+1}|\theta_{\mu'})|\theta_{Q'}) - \mathcal{Q}(s_t, a|\theta_\mathcal{Q}))^2] \tag{5}$$

## 4.2 ADAPTIVE IMPORTANCE-BASED REPLAY

The replay mechanism can help to alleviate the catastrophic forgetting (Borsos et al. (2020)). However, the operation of repeated replay easily causes the over-fitting problem, especially for time series with small difference between the two adjacent times. In CL, many methods only replay the representative data, such as the class means (Rebuffi et al. (2017)) and the class prototype (Mazumder et al. (2021)), each representative is fixed to its distribution. But in CCTS, we still need to consider whether all the representative need to be learned again and whether the representative will change over time.

Thus, we focus on the adaptive method to explore a wider space, where the replayed data is dynamic and determined according to the current state. We introduce an importance-based replay method. In each round, it only re-trained the model by some important samples to the model. The importance of each sample is learned from the objective of an additive loss function.

We incorporate the importance parameter $\beta_i$ of a time series $X_i$ in the replay buffer $\mathcal{B}^n$ as a coefficient of its loss $\mathcal{L}_{n,i}$. The overall loss at the current time $t^n$ is the sum of loss of each sample's loss:

$$\mathcal{L}_n = \frac{1}{|\mathcal{B}^n|}\sum_{i=1}^{|\mathcal{B}^n|}(\beta_{n,i}^2\mathcal{L}_{n,i} + \lambda(\beta_{n,i} - 1)^2) \tag{6}$$
$$\mathcal{B}^n = \{X_{1:t^n}, \tilde{X}_i | \beta_{n-1,i} < \epsilon\}$$

$\beta$ is learned by the gradient descent $\beta_{n,i} \leftarrow \beta_{n,i} - \frac{\partial\mathcal{L}_n}{\partial\beta_{n,i}}$. Thus, if a sample $X_i$ is hard to classify, its loss $\mathcal{L}_{*,i}$ will be larger. In order to minimize the loss, its $\beta_{*,i}$ will be smaller. Based on this, in each

---

**Algorithm 2** The Model Training Process by ACCTS Policy

---

**Input:** Data $\mathcal{D} = \{(X, C)\}$; ACCTS Actor $\mu$.  
**Output:** Final Classifier net $f^N$.

1: Initialize buffer $\mathcal{B}_1 \leftarrow \{X_{1:1}\}$  
2: Initialize DL Classifier net $f^1$.  
    //TRAVERSE EVERY TIME POINT  
3: **for** $t = 2$ to $T$ **do**  
4:    $s_t \leftarrow \{h_t, g_t\}$ from Eq.2  
5:    $a_t \leftarrow \mu(s_t)$

    //ADAPTIVE EXTRACTION  
6:   **if** $a_t = 1$ **then**  
7:     $\mathcal{B}_n \leftarrow \mathcal{B}_n + \{X_{1:t}\}$  
8:     $f^n \leftarrow$ Train $f^{n-1}$ by $\mathcal{B}^n$ with Eq.6  
      //IMPORTANCE STORAGE  
9:     $\mathcal{B}^{n+1} \leftarrow \{X_i | X_i \in \mathcal{B}^n, \beta_i < \epsilon\}$  
10:  **end if**  
11: **end for**

---

learning phrase, the buffer $\mathcal{B}^n$ contains the current time series $X_{1:t^n}$ and the important old time series $\tilde{X}$, who are the first few difficult learning samples ($\beta_{n-1,i} < \epsilon$) in the last buffer $\mathcal{B}^{n-1}$. Meanwhile, as $\beta$ is the confidence of loss, if $\beta = 0$, the loss are hard to be optimized. Thus, inspired by Kiyasseh et al. (2021), we introduce a regularization term $(\beta - 1)^2$ and initialize $\beta = 1$ to penalize it when rapidly decaying toward 0. As $\beta$ is re-obtained after each model training process, the important samples $\tilde{X}$ are changed adaptively and the buffer $\mathcal{B}$ is updated iteratively.

### 4.3 OVERALL TRAINING PROCESS FOR CLASSIFIER MODEL

The adaptive multi-distribution extraction policy, which is achieved by the Actor net $\mu$, is trained before the classifier training process, as shown in Algorithm 1. First, LSTM calculates the current sate $s_t$ (Line 2) and gives the action $a_t$ (Line 3). Then, the reward $r_t$ is obtained by the long-term accuracy to update the net (Line 6), where Actor and Critic are updated alternately. Main Critic net is updated by Q value, calculating from both two Critic. Main Actor is updated by the back-propagation gradient of main Critic. Target Actor and Critic are learned by the soft update (Line 7).

The adaptive importance-based replay policy is trained along with the classifier training process, as shown in Algorithm 2. First, in each time step, the Actor of ACCTS determines if a new distribution appears (Line 4,5). If yes, train the classifier from $f^n$ to $f^{n-1}$ by datasets in the buffer $\mathcal{B}^n$ (Line 7,8), and get the important samples according to $\beta$ to form a new buffer $\mathcal{B}^{n+1}$ (Line 9); Else, continue to get new values in next time point $t + 1$. At the final time, we can get the well-trained classifier $f^N$.

Note that the two processes of the adaptive multi-distribution extraction and the adaptive importance-based replay are relevant rather than independent. The extraction policy is based on the feature of the buffer data, and the replay policy selects the important samples based on the extracted data. Both of them are data-based, which helps for adaptive combination. That's why we design the replay-based policy rather than the regularization-based policy after the distribution extraction.

## 5 EXPERIMENTS

### 5.1 DATASETS

- UCR-EQ dataset (Chen et al. (2015)) has 471 earthquake records from UCR time series database archive. It is the univariate time series of seismic feature value. Natural disaster early warning, like earthquake warning, helps to reduce casualties and property losses (Ammon et al. (2021)).
- USHCN dataset (Menne & R. (2016)) has the daily meteorological data of 48 states in U.S. from 1887 to 2014. It is the multivariate time series of 5 weather features. Rainfall warning is not only the demand of daily life, but also can help prevent natural disasters (Lee et al. (2021)).
- COVID-19 dataset (Yan L (2020)) has 6,877 blood samples of 485 COVID-19 patients from Tongji Hospital, Wuhan, China. It is the multivariate time series of 74 laboratory test features. Mortality prediction helps for the personalized treatment and resource allocation (Sun et al. (2020)).
- SEPSIS dataset (Reyna et al. (2019b)) has 30,336 patients' records, including 2,359 diagnosed sepsis. It is the multivariate time series of 40 related patient features. Early diagnose of sepsis is critical to improve the outcome of ICU patients (Seymour et al. (2017)).

Not that for each time series in the above four datasets, every time point is tagged with a class label, which is the same as its outcome label, such as 'mortality', 'sepsis', 'earthquake' and 'rain'.

## 5.2 BASELINES

CCTS is related to ECTS and CL, we use baselines in these fields. The first is ECTS-based methods. All methods use the same base model of LSTM and the same structure of fully connected layers.

- LSTM (Wiens et al. (2012); Choi et al. (2017)). It contains a single classifier model. For one time series, the classifier model is trained by all subsequences from time 1 to time $t$, where $t = 2, ..., T$.
- SR (Mori et al. (2018)). It has multiple base models. All models are trained by the full-length time series. The final classification is the fusion result. It also has a stop rule of classification stop time.
- ECEC (Lv et al. (2019)). It has a set of base models. Each model is trained by time series in different time stages. When classifying, the data selects the classifier based on its time stages.

The second type is CL-based methods, including regularization and replay methods:

- EWC (Kirkpatrick et al. (2016)). It is a regularization-based method, training a model to remember the old tasks by constraining important parameters to stay close to their old values.
- GEM (Lopez-Paz & Ranzato (2017)). It is a regularization-based method, training a model to remember the old tasks by finding the new gradients which are at acute angles to the old gradients.
- CLEAR (Rolnick et al. (2019)). It is a replay-based method, using the reservoir sampling to limit the number of stored samples to a fixed budget assuming an i.i.d. data stream.
- CLOPS (Kiyasseh et al. (2021)). It is a replay-based method, training a base model by replaying old tasks with importance-guided buffer storage and uncertainty-based buffer acquisition.

## 5.3 EVALUATION METRICS

The accuracy is evaluated by Area Under Curve of Receiver Operating Characteristic (AUC-ROC). The performance of continuous mode is evaluated by Backward Transfer (BWT) and Forward Transfer (FWT), the influence that learning a current has on the old/future. $R \in \mathbb{R}^{|\mathcal{M}| \times |\mathcal{M}|}$ is an accuracy matrix, $R_{i,j}$ is the accuracy on $\mathcal{M}^j$ after learning $\mathcal{M}^i$. $\bar{b}$ is the accuracy with random initialization.

$$\text{BWT} = \frac{1}{|\mathcal{M}| - 1} \sum_{i=1}^{|\mathcal{M}|-1} R_{|\mathcal{M}|,i} - R_{i,i}, \quad \text{FWT} = \frac{1}{|\mathcal{M}| - 1} \sum_{i=2}^{|\mathcal{M}|} R_{i-1,i} - \bar{b}_{i,i} \qquad (7)$$

## 5.4 RESULTS AND ANALYSIS

We test the baselines from the classification accuracy, analyze our ACCTS methods from ablation study and coefficient test, and show the representation of time series in continuous classification.

### 5.4.1 CLASSIFICATION ACCURACY

ACCTS has the best performance on classification accuracy. As shown in Table 1, it can classify time series more accurately than all baselines at every time. The average accuracy is about 2% higher.

### 5.4.2 CATASTROPHIC FORGETTING AND OVER FITTING

ACCTS is the best when solving these two problems with the highest BWT and FWT as shown in Table 2. In Table 3, for CL-based methods (EWC, GEM, CLEAR and CLOPS), the accuracy on validation set is much lower than that on training set, shown as ↓. And in Table 1, in late stages, the accuracy of CL-based methods is also lower than that of ECTS-based methods. They all show the over-fitting of CL-based methods caused by training and reviewing at all time. But ACCTS can alleviate this problem by training and reviewing at the selected time so that it accuracy is higher.

### 5.4.3 ABLATION STUDY

Both policies of ACCTS are necessary. As shown in Figure 4, compared with the extraction at all times and random extraction, the adaptive extraction performs best in both overall data and the early distribution; Compared with no-replay and all-replay, the adaptive replay has the best performance. Besides, the accuracy of importance-based replay is higher than regularization. It demonstrate a good fit between two data-based polices of ACCTS: Importance-based replay and distribution extraction fit better than regularization and distribution extraction. This confirms the reason why we refer to replay-based instead of regularization-based CL methods.

### 5.4.4 COEFFICIENT TEST

ACCTS has two definable coefficients $\alpha$ and $\epsilon$, belong to two policies separately. Larger $\alpha$ review more distribution to learn. Larger $\epsilon$ causes more samples to review. As shown in Figure 5, the practice is to set them in the direct ratio: Within a reasonable range, more distributions need more review.

Table 1: Baselines Classification Accuracy (AUC-ROC↑) for 4 Real-world Datasets at 5 Time Steps.
*20% means the current classification time is 20% of the total time of the full-length time series;
Bold font indicates the highest accuracy;
More detailed results are in Appendix.

| Dataset | Method | 20%* | 40% | 60% | 80% | 100% |
|---|---|---|---|---|---|---|
| UCR-EQ | LSTM | 0.711±0.038 | 0.843±0.019 | 0.874±0.012 | 0.909±0.014 | 0.924±0.012 |
| | SR | 0.736±0.014 | 0.863±0.015 | 0.888±0.017 | 0.928±0.105 | 0.941±0.104 |
| | ECEC | 0.738±0.018 | 0.865±0.014 | 0.890±0.015 | 0.929±0.107 | 0.940±0.009 |
| | EWC | 0.768±0.018 | 0.874±0.016 | 0.895±0.014 | 0.923±0.102 | 0.933±0.003 |
| | GEM | 0.767±0.017 | 0.876±0.016 | .0900±0.015 | 0.929±0.008 | 0.934±0.004 |
| | CLEAR | 0.770±0.015 | 0.880±0.013 | 0.904±0.012 | 0.923±0.004 | 0.932±0.005 |
| | CLOPS | 0.773±0.016 | 0.878±0.016 | 0.902±0.015 | 0.917±0.006 | 0.925±0.005 |
| | **ACCTS** | **0.774±0.023** | **0.882±0.022** | **0.906±0.005** | **0.933±0.010** | **0.946±0.003** |
| USHCN | LSTM | 0.700±0.028 | 0.745±0.028 | 0.820±0.015 | 0.852±0.014 | 0.891±0.002 |
| | SR | 0.730±0.022 | 0.761±0.023 | 0.836±0.016 | 0.902±0.013 | 0.933±0.009 |
| | ECEC | 0.736±0.024 | 0.760±0.025 | 0.837±0.016 | 0.906±0.017 | 0.931±0.009 |
| | EWC | 0.736±0.025 | 0.798±0.024 | 0.834±0.016 | 0.896±0.017 | 0.926±0.007 |
| | GEM | 0.728±0.026 | 0.781±0.023 | 0.838±0.013 | 0.899±0.010 | 0.928±0.005 |
| | CLEAR | 0.738±0.025 | 0.784±0.024 | 0.837±0.010 | 0.879±0.012 | 0.921±0.004 |
| | CLOPS | 0.740±0.024 | 0.781±0.025 | 0.835±0.016 | 0.877±0.011 | 0.919±0.013 |
| | **ACCTS** | **0.742±0.017** | **0.791±0.021** | **0.841±0.012** | **0.910±0.015** | **0.939±0.013** |
| COVID-19 | LSTM | 0.701±0.033 | 0.833±0.015 | 0.888±0.013 | 0.925±0.014 | 0.944±0.015 |
| | SR | 0.730±0.024 | 0.867±0.016 | 0.900±0.018 | 0.946±0.006 | 0.962±0.005 |
| | ECEC | 0.732±0.028 | 0.870±0.016 | 0.904±0.014 | 0.948±0.015 | 0.963±0.017 |
| | EWC | 0.769±0.015 | 0.888±0.028 | 0.923±0.014 | 0.940±0.013 | 0.954±0.008 |
| | GEM | 0.779±0.017 | 0.885±0.022 | 0.924±0.018 | 0.939±0.010 | 0.953±0.005 |
| | CLEAR | 0.785±0.019 | 0.879±0.016 | 0.926±0.014 | 0.941±0.007 | 0.952±0.008 |
| | CLOPS | 0.775±0.013 | 0.900±0.017 | 0.925±0.015 | 0.940±0.007 | 0.954±0.006 |
| | **ACCTS** | **0.790±0.023** | **0.901±0.022** | **0.927±0.006** | **0.960±0.011** | **0.967±0.008** |
| SEPSIS | LSTM | 0.629±0.035 | 0.736±0.064 | 0.748±0.043 | 0.795±0.027 | 0.827±0.039 |
| | SR | 0.659±0.015 | 0.791±0.026 | 0.827±0.037 | 0.845±0.014 | 0.866±0.023 |
| | ECEC | 0.669±0.019 | 0.793±0.016 | 0.815±0.014 | 0.849±0.016 | 0.863±0.014 |
| | EWC | 0.733±0.023 | 0.827±0.036 | 0.838±0.024 | 0.848±0.015 | 0.854±0.016 |
| | GEM | 0.730±0.024 | 0.826±0.033 | 0.836±0.028 | 0.849±0.014 | 0.853±0.012 |
| | CLEAR | 0.732±0.024 | 0.825±0.035 | 0.839±0.028 | 0.847±0.010 | 0.848±0.016 |
| | CLOPS | 0.733±0.025 | 0.824±0.036 | 0.838±0.026 | 0.850±0.017 | 0.857±0.018 |
| | **ACCTS** | **0.734±0.038** | **0.828±0.030** | **0.842±0.034** | **0.857±0.012** | **0.872±0.012** |

Table 2: Continual Learning Performance of Baselines.
The left table is BWT↑ results, the right table is FWT↑ results.

| Dataset \ Method | EWC | GEM | CLEAR | CLOPS | **ACCTS** |
|---|---|---|---|---|---|
| UCR-EQ | +0.039 | +0.041 | +0.053 | +0.052 | **+0.058** |
| USHCN | +0.058 | +0.054 | +0.063 | +0.074 | **+0.084** |
| COVID-19 | +0.011 | +0.012 | +0.009 | +0.014 | **+0.020** |
| SEPSIS | +0.019 | +0.017 | +0.030 | +0.032 | **+0.035** |

| Dataset \ Method | EWC | GEM | CLEAR | CLOPS | **ACCTS** |
|---|---|---|---|---|---|
| UCR-EQ | +0.321 | +0.329 | +0.312 | +0.301 | **+0.345** |
| USHCN | +0.312 | +0.328 | +0.335 | +0.301 | **+0.342** |
| COVID-19 | +0.426 | +0.421 | +0.427 | +0.439 | **+0.455** |
| SEPSIS | +0.295 | +0.265 | +0.401 | +0.397 | **+0.410** |

Table 3: COVID-19 Classification Accuracy with Non-uniform Training Sets and Validation Sets.
↓ means the accuracy is greatly reduced; More detailed results are in the appendix.

| Subset | SR | ECEC | EWC | GEM | CLEAR | CLOPS | ACCTS |
|---|---|---|---|---|---|---|---|
| Male | 0.968±0.014 | 0.969±0.016 | 0.965±0.012 | 0.965±0.004 | 0.978±0.009 | 0.978±0.014 | 0.971±0.010 |
| Female | 0.945±0.004 | 0.947±0.015 | 0.939±0.018 | 0.938±0.003 | 0.919±0.008↓ | 0.921±0.009↓ | 0.947±0.002 |
| Age 30- | 0.965±0.014 | 0.967±0.015 | 0.967±0.013 | 0.964±0.009 | 0.977±0.008 | 0.979±0.012 | 0.972±0.010 |
| Age 30+ | 0.941±0.007 | 0.943±0.018 | 0.931±0.008↓ | 0.923±0.040↓ | 0.902±0.006↓ | 0.914±0.007↓ | 0.945±0.006 |
| Test | 0.964±0.013 | 0.968±0.015 | 0.966±0.012 | 0.962±0.006 | 0.979±0.009 | 0.978±0.010 | 0.970±0.007 |
| Valid. | 0.962±0.006 | 0.963±0.014 | 0.954±0.003 | 0.953±0.005 | 0.952±0.009↓ | 0.954±0.004↓ | 0.967±0.006 |

Figure 4: Ablation Study of Two Policies of ACCTS with the Case Study of COVID-19

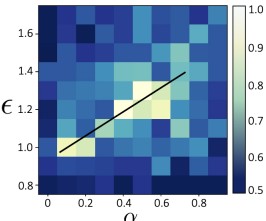

Figure 5: Parameter Test of $\alpha, \epsilon$ in ACCTS.

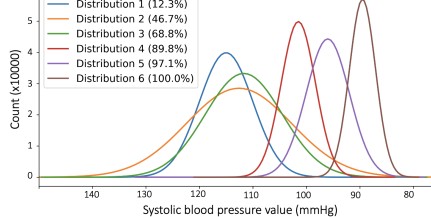

Figure 6: Extracted Six Distributions from SEPSIS Dataset by ACCTS.

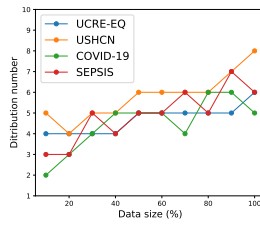

Figure 7: Impact of Data Size to Distributions.

Figure 8: The Important Samples in Four SEPSIS Distribution Buffers (2,3,4,5 in Figure 6)

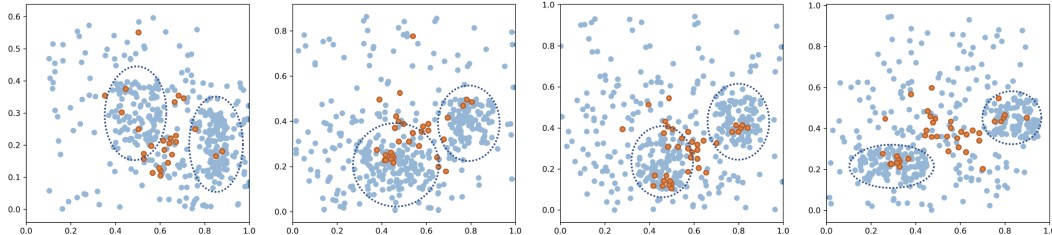

### 5.4.5 ANALYSIS OF MULTI-DISTRIBUTION

The case study of SEPSIS in Figure 6 shows that ACCTS only extracts six distributions and the difference among distributions is relatively large. The extraction is concentrated in 85%-length late stage, which may be because the patient's vital signs change significantly near the outcome time.

### 5.4.6 ANALYSIS OF IMPORTANT SAMPLES

The important samples includes not only the data hard to learn, but also the representative data as shown in Figure 8. It might because that, the representative data is similar to the most common data, resulting in a greater additive loss, therefore leading to smaller coefficients in Equation 6. Through the experimental results, we can redefine the important samples: Important samples are samples that can represent most data of a class and samples that are difficult to distinguish by the model.

### 5.4.7 IMPACT OF DATA SIZE

ACCTS prefers to divide more distributions for large data size as shown in Figure 7. Besides, ACCTS has greater advantages over baselines in larger data size, detailed results are in Appendix.

## 6 CONCLUSION

In this paper, we propose a new concept of Continuous Classification of Time Series (CCTS) to meet the real needs. It has two major difficulties of catastrophic forgetting and over fitting. In CCTS, the multi-distribution of time series is not clearly defined, and the distribution division directly affects the above two difficulties. Thus, we design an Adaptive model training policy named ACCTS. It contains a multi-distribution extraction policy adaptive to the time series evaluation and the model change, and an importance-based replay policy adaptive to the data features and final accuracy. We test the methods on four real-world datasets and analyze the method from perspectives of accuracy, continual learning, ablation study and parameter setting. Future work could refer to the distribution extraction policy that fits to regularization-based methods.

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

# A DETAILED EXPERIMENTAL RESULTS

## A.1 BASIC CLASSIFICATION ACCURACY

Table 4: Classification Accuracy (AUC-ROC↑) for 4 Real-world Datasets at the First 5 Time Steps.
*k% means the current classification time is k% of the total time of the full-length time series;
Bold font indicates the highest accuracy.

| Dataset | Method | 10% | 20% | 30% | 40% | 50% |
|---------|--------|-----|-----|-----|-----|-----|
| UCR-EQ | LSTM | 0.695±0.044 | 0.711±0.038 | 0.803±0.024 | 0.843±0.019 | 0.854±0.017 |
| | SR | 0.700±0.015 | 0.736±0.014 | 0.830±0.016 | 0.863±0.015 | 0.871±0.024 |
| | ECEC | 0.703±0.013 | 0.738±0.018 | 0.828±0.017 | 0.865±0.014 | 0.873±0.026 |
| | EWC | 0.724±0.015 | 0.768±0.018 | 0.848±0.014 | 0.874±0.016 | 0.883±0.025 |
| | GEM | 0.723±0.014 | 0.767±0.017 | 0.850±0.015 | 0.876±0.016 | 0.890±0.024 |
| | CLEAR | 0.729±0.015 | 0.770±0.015 | 0.852±0.019 | 0.880±0.013 | 0.899±0.026 |
| | CLOPS | 0.728±0.016 | 0.773±0.016 | 0.855±0.015 | 0.878±0.016 | 0.896±0.028 |
| | **ACCTS** | **0.730±0.022** | **0.774±0.023** | **0.856±0.015** | **0.882±0.022** | **0.900±.0017** |
| USHCN | LSTM | 0.682±0.014 | 0.700±0.028 | 0.721±0.013 | 0.745±0.028 | 0.784±0.023 |
| | SR | 0.702±0.014 | 0.730±0.022 | 0.745±0.016 | 0.761±0.023 | 0.809±0.024 |
| | ECEC | 0.707±0.017 | 0.736±0.024 | 0.748±0.015 | 0.760±0.025 | 0.806±0.025 |
| | EWC | 0.727±0.018 | 0.736±0.025 | 0.768±0.017 | 0.798±0.024 | 0.805±0.022 |
| | GEM | 0.720±0.019 | 0.728±0.026 | 0.772±0.015 | 0.781±0.023 | 0.801±0.026 |
| | CLEAR | 0.728±0.016 | 0.738±0.025 | 0.773±0.018 | 0.784±0.024 | 0.802±0.027 |
| | CLOPS | 0.728±0.012 | 0.740±0.024 | 0.769±0.019 | 0.781±0.025 | 0.800±0.024 |
| | **ACCTS** | **0.730±0.018** | **0.742±0.017** | **0.775±0.016** | **0.791±0.021** | **0.810±.0133** |
| COVID-19 | LSTM | 0.605±0.044 | 0.701±0.033 | 0.793±0.022 | 0.833±0.015 | 0.844±0.013 |
| | SR | 0.636±0.014 | 0.730±0.024 | 0.810±0.013 | 0.867±0.016 | 0.901±0.013 |
| | ECEC | 0.639±0.013 | 0.732±0.028 | 0.829±0.013 | 0.870±0.016 | 0.901±0.026 |
| | EWC | 0.703±0.022 | 0.769±0.015 | 0.870±0.014 | 0.888±0.028 | 0.915±0.017 |
| | GEM | 0.699±0.025 | 0.779±0.017 | 0.871±0.015 | 0.885±0.022 | 0.914±0.019 |
| | CLEAR | 0.710±0.013 | 0.785±0.019 | 0.870±0.016 | 0.879±0.016 | 0.916±0.024 |
| | CLOPS | 0.709±0.017 | 0.775±0.013 | 0.869±0.012 | 0.900±0.017 | 0.918±0.026 |
| | **ACCTS** | **0.712±0.021** | **0.790±0.023** | **0.872±0.013** | **0.901±0.022** | **0.919±0.016** |
| SEPSIS | LSTM | 0.576±0.063 | 0.629±0.035 | 0.735±0.064 | 0.736±0.064 | 0.745±0.056 |
| | SR | 0.626±0.035 | 0.659±0.015 | 0.768±0.013 | 0.791±0.026 | 0.803±0.018 |
| | ECEC | 0.623±0.024 | 0.669±0.019 | 0.761±0.016 | 0.793±0.016 | 0.811±0.015 |
| | EWC | 0.671±0.027 | 0.733±0.023 | 0.799±0.015 | 0.827±0.036 | 0.832±0.028 |
| | GEM | 0.670±0.026 | 0.730±0.024 | 0.802±0.018 | 0.826±0.033 | 0.834±0.026 |
| | CLEAR | 0.680±0.028 | 0.732±0.024 | 0.801±0.015 | 0.825±0.035 | 0.833±0.025 |
| | CLOPS | 0.684±0.025 | 0.733±0.025 | 0.802±0.017 | 0.824±0.036 | 0.830±0.023 |
| | **ACCTS** | **0.690±0.032** | **0.734±0.038** | **0.812±0.022** | **0.828±0.036** | **0.835±0.024** |

Table 5: Classification Accuracy (AUC-ROC↑) for 4 Real-world Datasets at the Last 5 Time Steps.
*k% means the current classification time is k% of the total time of the full-length time series;
Bold font indicates the highest accuracy.

| Dataset | Method | 60% | 70% | 80% | 90% | 100% |
|---|---|---|---|---|---|---|
| UCR-EQ | LSTM | 0.874±0.012 | 0.913±0.034 | 0.909±0.014 | 0.919±0.008 | 0.924±0.012 |
| | SR | 0.888±0.017 | 0.924±0.010 | 0.928±0.105 | 0.936±0.103 | 0.941±0.104 |
| | ECEC | 0.890±0.015 | 0.923±0.013 | 0.929±0.107 | 0.936±0.006 | 0.940±0.009 |
| | EWC | 0.895±0.014 | 0.910±0.017 | 0.923±0.102 | 0.930±0.005 | 0.933±0.003 |
| | GEM | 0.900±0.015 | 0.920±0.015 | 0.929±0.008 | 0.935±0.003 | 0.934±0.004 |
| | CLEAR | 0.904±0.012 | 0.918±0.019 | 0.923±0.004 | 0.928±0.007 | 0.932±0.005 |
| | CLOPS | 0.902±0.015 | 0.915±0.010 | 0.917±0.006 | 0.921±0.009 | 0.925±0.005 |
| | **ACCTS** | **0.906±0.005** | **0.928±0.007** | **0.933±0.010** | **0.940±0.005** | **0.946±0.003** |
| USHCN | LSTM | 0.820±0.015 | 0.837±0.024 | 0.852±0.014 | 0.869±0.025 | 0.891±0.002 |
| | SR | 0.836±0.016 | 0.886±0.023 | 0.902±0.013 | 0.921±0.026 | 0.933±0.009 |
| | ECEC | 0.837±0.016 | 0.887±0.027 | 0.906±0.017 | 0.920±0.028 | 0.931±0.009 |
| | EWC | 0.834±0.016 | 0.867±0.026 | 0.896±0.017 | 0.906±0.020 | 0.926±0.007 |
| | GEM | 0.838±0.013 | 0.868±0.029 | 0.899±0.010 | 0.910±0.021 | 0.928±0.005 |
| | CLEAR | 0.837±0.010 | 0.867±0.023 | 0.879±0.012 | 0.899±0.027 | 0.921±0.004 |
| | CLOPS | 0.835±0.016 | 0.861±0.024 | 0.877±0.011 | 0.895±0.016 | 0.919±0.013 |
| | **ACCTS** | **0.841±0.012** | **0.898±0.022** | **0.910±0.015** | **0.928±0.013** | **0.939±0.013** |
| COVID-19 | LSTM | 0.888±0.013 | 0.918±0.033 | 0.925±0.014 | 0.939±0.005 | 0.944±0.015 |
| | SR | 0.900±0.018 | 0.935±0.010 | 0.946±0.006 | 0.952±0.017 | 0.962±0.005 |
| | ECEC | 0.904±0.014 | 0.937±0.008 | 0.948±0.015 | 0.952±0.008 | 0.963±0.017 |
| | EWC | 0.923±0.014 | 0.935±0.007 | 0.940±0.013 | 0.950±0.013 | 0.954±0.008 |
| | GEM | 0.924±0.018 | 0.936±0.009 | 0.939±0.010 | 0.949±0.017 | 0.953±0.005 |
| | CLEAR | 0.926±0.014 | 0.933±0.011 | 0.941±0.007 | 0.948±0.009 | 0.952±0.008 |
| | CLOPS | 0.925±0.015 | 0.935±0.013 | 0.940±0.007 | 0.947±0.006 | 0.954±0.006 |
| | **ACCTS** | **0.927±0.006** | **0.955±0.008** | **0.960±0.011** | **0.963±0.009** | **0.967±0.008** |
| SEPSIS | LSTM | 0.748±0.043 | 0.773±0.032 | 0.795±0.027 | 0.813±0.025 | 0.827±0.039 |
| | SR | 0.827±0.037 | 0.835±0.013 | 0.845±0.014 | 0.859±0.022 | 0.866±0.023 |
| | ECEC | 0.815±0.014 | 0.827±0.016 | 0.849±0.016 | 0.859±0.017 | 0.863±0.014 |
| | EWC | 0.838±0.024 | 0.842±0.030 | 0.848±0.017 | 0.850±0.014 | 0.854±0.016 |
| | GEM | 0.836±0.028 | 0.841±0.034 | 0.849±0.014 | 0.851±0.016 | 0.853±0.012 |
| | CLEAR | 0.839±0.028 | 0.842±0.031 | 0.847±0.010 | 0.850±0.019 | 0.848±0.016 |
| | CLOPS | 0.838±0.026 | 0.842±0.030 | 0.850±0.017 | 0.853±0.010 | 0.857±0.018 |
| | **ACCTS** | **0.842±0.034** | **0.852±0.023** | **0.857±0.012** | **0.866±0.014** | **0.872±0.012** |

## A.2 CLASSIFICATION ACCURACY ON SUBSETS WITH DIFFERENT SEMANTICS

Table 6: COVID-19 Classification Accuracy with Non-uniform Training Sets and Validation Sets. ↓ means the accuracy is greatly reduced.

| Subset | LSTM | SR | ECEC | EWC |
|---|---|---|---|---|
| Male | 0.955±0.013 | 0.968±0.014 | 0.969±0.016 | 0.965±0.012 |
| Female | 0.924±0.013 | 0.945±0.004 | 0.947±0.015 | 0.939±0.018 |
| Age 30- | 0.954±0.013 | 0.965±0.014 | 0.967±0.015 | 0.967±0.013 |
| Age 30+ | 0.923±0.014 | 0.941±0.007 | 0.943±0.018 | 0.931±0.008↓ |
| Test | 0.950±0.011 | 0.964±0.013 | 0.968±0.015 | 0.966±0.012 |
| Valid. | 0.944±0.014 | 0.962±0.006 | 0.963±0.014 | 0.954±0.003 |
| Subset | GEM | CLEAR | CLOPS | ACCTS |
| Male | 0.965±0.004 | 0.978±0.009 | 0.978±0.014 | 0.971±0.010 |
| Female | 0.938±0.003 | 0.919±0.008↓ | 0.921±0.009↓ | 0.947±0.002 |
| Age 30- | 0.964±0.009 | 0.977±0.008 | 0.979±0.012 | 0.972±0.010 |
| Age 30+ | 0.923±0.040↓ | 0.902±0.006↓ | 0.914±0.007↓ | 0.945±0.006 |
| Test | 0.962±0.006 | 0.979±0.009 | 0.978±0.010 | 0.970±0.007 |
| Valid. | 0.953±0.005 | 0.952±0.009↓ | 0.954±0.004↓ | 0.967±0.006 |

A.3 CLASSIFICATION ACCURACY ON SUBSETS WITH DIFFERENT DATA SIZE

Table 7: Classification Accuracy (AUC-ROC↑) for Subsets with Different Data Size.
k% means the volume of sub dataset is k% of the corresponding original dataset;
Bold font indicates the highest accuracy;
* means that the accuracy of ACCTS is higher 2% than this method.

| Dataset | Method | 20% | 40% | 60% | 80% | 100% |
|---------|--------|------|------|------|------|------|
| UCR-EQ | LSTM | 0.724* | 0.765* | 0.804* | 0.809* | 0.813* |
| | SR | 0.758* | 0.784* | 0.828* | 0.813* | 0.831* |
| | ECEC | 0.790 | 0.770* | 0.815* | 0.827* | 0.838* |
| | EWC | 0.785 | 0.791* | 0.833* | 0.855* | 0.862* |
| | GEM | 0.780 | 0.775* | 0.840* | 0.857* | 0.863* |
| | CLEAR | 0.784 | 0.808 | 0.859 | 0.864* | 0.870* |
| | CLOPS | 0.792 | 0.809 | 0.864 | 0.871 | 0.875* |
| | **ACCTS** | **0.797** | **0.817** | **0.872** | **0.886** | **0.896** |
| USHCN | LSTM | 0.701* | 0.730* | 0.732* | 0.760* | 0.763* |
| | SR | 0.731* | 0.769* | 0.782* | 0.801* | 0.805* |
| | ECEC | 0.747* | 0.774 | 0.800* | 0.807* | 0.816* |
| | EWC | 0.739* | 0.768* | 0.810 | 0.817 | 0.826* |
| | GEM | 0.737* | 0.772 | 0.809 | 0.811* | 0.818* |
| | CLEAR | 0.757 | 0.780 | 0.812 | 0.819 | 0.823* |
| | CLOPS | 0.775 | 0.785 | 0.817 | 0.825 | 0.839 |
| | **ACCTS** | **0.776** | **0.790** | **0.821** | **0.835** | **0.843** |
| COVID-19 | LSTM | 0.713* | 0.730* | 0.765* | 0.819* | 0.834* |
| | SR | 0.751* | 0.767* | 0.806 | 0.822* | 0.842* |
| | ECEC | 0.755* | 0.770* | 0.796* | 0.829* | 0.856* |
| | EWC | 0.763 | 0.785 | 0.794* | 0.835* | 0.849* |
| | GEM | 0.769 | 0.772* | 0.793* | 0.849 | 0.856* |
| | CLEAR | 0.776 | 0.791 | 0.810 | 0.856 | 0.866* |
| | CLOPS | 0.775 | 0.789 | 0.809 | 0.848 | 0.874 |
| | **ACCTS** | **0.781** | **0.800** | **0.821** | **0.863** | **0.888** |
| SEPSIS | LSTM | 0.658* | 0.669* | 0.691* | 0.733* | 0.747 |
| | SR | 0.682 | 0.700 | 0.725* | 0.759* | 0.768 |
| | ECEC | 0.679* | 0.702 | 0.719* | 0.755* | 0.770 |
| | EWC | 0.685 | 0.708 | 0.729* | 0.768* | 0.772* |
| | GEM | 0.693 | 0.704 | 0.740* | 0.771* | 0.781* |
| | CLEAR | 0.687 | 0.705 | 0.741 | 0.776 | 0.789 |
| | CLOPS | 0.698 | 0.710 | 0.745 | 0.779 | 0.783* |
| | **ACCTS** | **0.701** | **0.712** | **0.760** | **0.794** | **0.803** |

