# OpenReview forum: "ACCTS: an Adaptive Model Training Policy for Continuous Classification of Time Series"
_ICLR.cc/2022/Conference — ICLR 2022 Submitted_

### Official Review · Reviewer_Kj5t · 2021-10-20

**Correctness:** 2
**Technical Novelty And Significance:** 3
**Empirical Novelty And Significance:** 2
**Recommendation:** 3
**Confidence:** 2

**Main Review:**

Here are some of my concerns regards to this manuscript.

#### <Problem definition>
Isn't the continuous classification you are talking about the same as forecasting?
Any real-world time-series forecasting application handles this problem, and there are many services and implementations to process time-series streams.
(e.g., https://cloud.google.com/architecture/processing-streaming-time-series-data-overview, https://d1.awsstatic.com/whitepapers/time-series-forecasting-principles-amazon-forecast.pdf, https://github.com/AIStream-Peelout/flow-forecast). If we do the aforementioned early classification (= point forecast) every time we receive new data, that becomes continuous time series classification (=forecasting with data streams). Alternatively, you may want to forecast in a long range, such as weather forecast. There are also lots of works on long-term, or long-range forecasting. So can you tell us a little more about which part is the new problem? Besides, the writing and organization of the paper have a large space for improvement. In its current version, the discussion of basic concept and ideas look fragmented and incoherent.

#### <Presentation>
It is not easy to read through the paper to understand how the ACCTS model works. For example, when you apply a method (e.g., Actor-Critic), please justify why this method is used. Another example is that I could not follow the logic in the first paragraph of Section 4.2 on why the importance-based replay method is needed.
Although I did not understand it fully, the idea of determining the right time to train the model and the proposed RL-based solution seem to have some technical merits, so I hope that the authors can spend more effort clarifying their ideas and make their writing coherent so that readers can have a better experience reading it. Also, avoid using vague terms like “we focus on the adaptive method to explore a wider space" without clearly elaborating on its meaning.

#### <Evaluation>
Plus, it is insufficient to evaluate the algorithms only on classification accuracy. Survival analysis would be a relevant area to discuss COVID-19 and SEPSIS datasets, so please consider the metrics used in survival analysis.


#### <Minor issues>
Eq. 1 RHS: Should summation be done from t=1 to t^{n-1}?
p. 4: What do MG stand for?
p. 4: polity -> policy
p. 5: What do you mean by less divisions?
Eq. 7: Is BWT defined correctly?


**Summary Of The Paper:**

This paper proposes an adaptive model training policy (ACCTS) for continuous classification of time series (CCTS). CCTS is a problem setup that aims to forecast the target variable every time. To mitigate catastrophic forgetting and overfitting problem in CCTS, this paper defines a continual learning task and introduces ACCTS. ACCTS composes of three steps. The first step aims to decide whether to extract the current time-series sequence to train the model. The second step uses an adaptive method to explore a wider space and uses importance-based replay to alleviate catastrophic forgetting. These two processes are relevant to each other and aim to optimize long-term rewards. ACCTS is compared with seven baselines representing early-classification time series and continual learning on four datasets.

**Summary Of The Review:**

This paper proposes an adaptive solution for sustainable time-series forecasting. Although it seems to have some technical merits, the presentation needs to be improved a lot, and comprehensive evaluations will be required.

---

### Official Review · Reviewer_kGRM · 2021-10-31

**Correctness:** 3
**Technical Novelty And Significance:** 2
**Empirical Novelty And Significance:** 3
**Recommendation:** 3
**Confidence:** 3

**Main Review:**

positives:
- This paper studies a critical problem in real-world applications. The experimental results look promising and show the effectiveness of their proposed method.

negatives:
- Many notations are used without formal definition, making this work hard to follow. For example, in section 4, many notations are confusing to me.
- The application scope of the proposed method is limited. The proposed method is time-consuming as it retrains on selected replicas of received data at each time and needs a large buffer for data storage. Thus, it is only feasible in some specific applications where the accumulated data is small and not time-demanding for inference. However, in many real-world applications of streaming learning, such as the online product recommendation, a large amount of data are accumulating in a short period, and we need to change the model for inference adaptively.
- This draft lacks some essential discussions on a related topic of online learning in non-stationary environments, see [1] and follow-up works.
- The performance of the distribution extraction (segmentation) method seems to lack experimental verification. It is suggested to show the distribution change of the whole data stream, and empirically studies whether the segmentation point identifies the distribution change point.


[1] Besbes, Omar, Yonatan Gur, and Assaf Zeevi. "Non-stationary stochastic optimization." Operations research 63.5 (2015): 1227-1244.

**Summary Of The Paper:**

This draft attempts a streaming classification approach that adaptively identifies the distribution change and retrains the received data to perform well on the whole time series. The proposed method is based on a time series segmentation mechanism inspired by the Markov decision process and a retraining procedure that replays and retrain the selected samples. The empirical evaluation is performed on several real-world datasets.

**Summary Of The Review:**

This draft proposes a heuristic approach for streaming data classification. The method stores some selected replicas of received data and retrains the model at each time, which could hardly handle real-world streaming data.

---

### Official Review · Reviewer_Fizc · 2021-11-13

**Correctness:** 3
**Technical Novelty And Significance:** 3
**Empirical Novelty And Significance:** 2
**Recommendation:** 3
**Confidence:** 3

**Main Review:**

Pros:
- I think the application of RL to ECTS is novel and interesting.
- The experimental results look good and could potentially show the effectiveness of this approach.

Cons:
- First and foremost, in the current form, the draft is hard to read. I believe it's a combination of fragmented sentences within paragraphs, the usage of notations, and the lack of justification for certain modeling choices or evaluation metrics.
- Although the experimental results look promising, the baselines are quite lacking and I'm not convinced of the result. Since the tasks are downstream classification tasks, I would want to see more SOTA time-series classification methods besides just LSTM, SR, and ECEC. I understand that it might be prohibitive to train time-series models on every timestep, but in the paper, evaluations are carried out at 5 checkpoints (20,40,60,80,100) so I think we should be able to train more methods here. See [1] for a strong time-series baseline and other methods in their paper.
- Since the model requires retraining, I'm not sure if there's a runtime benefit over training multi-models by time-splitting. Even so, besides "the numbers look better", are there any insights on why this approach can better model the changing distributions? More concretely,  it would be nice to have a toy dataset with known distribution change or if the authors can show the distribution changes of the 4 real-world used in the evaluation, and empirically study how the approach identities when the distribution changes.


-- Minor --
Definition 1, do the label C varies with time or is it just one outcome label for every timestep? p. 4: What is MG? model gradient?

[1] Fawaz, Hassan Ismail, et al. "Inceptiontime: Finding alexnet for time series classification." Data Mining and Knowledge Discovery 34.6 (2020): 1936-1962.

**Summary Of The Paper:**

This work proposes to continuously classify time-series at every timestep to provide early-classification capability and address real-world needs. The major challenges of this task are catastrophic forgetting and overfitting. The authors state that the multi-distribution of time-series is not clearly defined and the different distributions cause problems with forgetting or overfitting. They proposed an adaptive model training policy to extract multi-distribution instead of rules and priors and an importance-based reply component to replay important experience. They tested on four real-world datasets.

**Summary Of The Review:**

The paper proposed a reinforcement learning method for early classification of time series. Although it seems to have some technical novelty, the presentation and evaluation will need to be improved.

---

### Official Review · Reviewer_AmKR · 2021-11-16

**Correctness:** 2
**Technical Novelty And Significance:** 2
**Empirical Novelty And Significance:** 3
**Recommendation:** 3
**Confidence:** 3

**Main Review:**

**Problem:**

The authors address the problem of generating a class prediction $c_t$ at each time point $t$, given observations $x_{1:t}$ made thus far. In particular, it is supposed that the true class may change over time, so that this problem is not just equivalent to early classification or time series classification from a subsequence.

The claim ($\S 1$, $\S 2.2$) that this problem setting is novel is not supported and warrants some scrutiny. How is it different from estimating the filtering distribution $p(y_t | x_{1:t})$ in a state-space model with discrete latent states and continuous observations? How is it different from a standard sequence-to-sequence learning problem, in which the target sequence takes values in the label space? Note that the case in which the number of classes is unknown or grows over time is well-known, see e.g. [1].

**Presentation:**

There is a lack of clarity in the presentation of this work that goes beyond some superficial writing errors. Several terms are non-standard but not defined; e.g. "multi-distribution" and "multi-model" ($\S 2.2$).

The mathematical notation is inconsistent and unclear. In Definitions 1 and 2, the class $C$ does not vary with time. The "distributions" $\mathcal{M}^n$ in Definition 2 are not defined. The loss function $\mathcal{L}$ takes different arguments in Eq (1) than elsewhere in the preceding definitions. In Algorithm 1, it is not clear how the "update" in line 6 makes use of the gradients, and $\mathcal{Q}', \mu'$ are not defined.

Some claims made in the paper are simply incorrect as stated:
- On page 1, the authors write: "However, a single model, like deep neural network, is lack of ability to learn all distributions simultaneously as they are restricted by the premise of independent identically distributed (i.i.d) data (Shim et al. (2021))." In reality, a large and well-known class of deep neural nets - RNNs - are explicitly designed to model non i.i.d. data, specifically time series. The authors use such models as baselines and as part of their own method.
- On page 3, the authors write: "Both CTS and ECTS are belong to this and always use Deep Learning (DL) models." Time series classification (as distinguished from class prediction per time point) significantly predates deep learning. There exist, for example, an extensive variety of kernel-based or clustering methods.

**Results:**

The empirical results show that the proposed method results in improved class prediction accuracy, in the sense of AUC, at each of several time points across a variety of datasets. In addition, the authors include useful analyses of backward and forward transfer that specifically address the question of successful continual learning, as well as a variety of ablation or parameter sensitivity experiments to study their particular solution. The work here is extensive and the results appear promising. Perhaps due to space constraints, there is relatively little discussion of the results presented in Figures 4-8; they are not even captioned.

A major issue with this section is that there is almost no information provided on how the proposed model was trained, for any experiment. There is no documentation of model architectures, hyperparameter settings, optimization procedures, or convergence criteria. It does not seem possible that a competent scientist could reproduce these results, even with unlimited resources.

There is also a question of computational complexity. The authors propose an RL-based method to adaptively segment the time series data, and then to repeatedly re-train the classification model on old segments in addition to the latest segment. There is no discussion of how long this takes relative to the baselines, but it is reasonable to suspect that the proposed method may be significantly slower.

**Other comments:**
- The shapelets reference on p. 2 is not correct; the method is due to Ye and Keogh (2009).

- The accuracy results are stated as $AUC \pm x$, but it is unclear what $x$ is. Where does the variability in the result come from?

- It seems that all of the data examples are limited to binary classification. Moreover, some seem a little dubious for the "continuous classification" setting - for example, mortality is a label that starts at zero, switches to one once, and then stays at one. It would be more reasonable to model this in a time-to-event framework than as continuous classification.

References:

[1] Fox, E. B., Sudderth, E. B., Jordan, M. I., & Willsky, A. S. (2011). A sticky HDP-HMM with application to speaker diarization. The Annals of Applied Statistics, 1020-1056.

**Summary Of The Paper:**

The authors consider a classification setting in which a class label $c_t$ is to be predicted on the basis of time series observations $(x_1, ..., x_t)$ up to time $t$. The goal is to maintain accurate prediction even as the generating distribution evolves, while retaining good accuracy on (i.e. not "forgetting" about) previously observed data. They propose a Markov decision process model to segment the time series, followed by a replay-based procedure in which the classification model is updated according to both the current segment and a collection of past segments. The method is contrasted against some baselines on four time series datasets.

**Summary Of The Review:**

This paper proposes a complex solution for what is claimed to be a novel problem, but the problem setting as stated seems well-known and the solution is unclear both in its motivation and its details. The results appear promising but lack discussion, and the experiments are poorly documented. My conclusion is that the paper is not ready for publication at this time.

---

### Decision · Program_Chairs · 2022-01-20

**Decision:**

Reject

**Comment:**

This paper presents a reinforcement learning algorithm to target variable in every time step.  Although the paper proposes an important problem in many real-world applications, there were various major criticisms raised by reviewers.  Most importantly, technical novelty is not well motivated or justified.  There is also a significant lack of a specific description of the proposed method, discussion of computational complexity, clarity and presentation, and evaluation metrics, which decreased the enthusiasm of the reviewers.